Identification and expression analysis of maize NF-YA subunit genes

Lv Mingyue 1 2
http://orcid.org/0000-0001-9244-583X Cao Hongzhe 1 2
Wang Xue 2
Zhang Kang 1 2
Si Helong 1 2
Zang Jinping 1 2
Xing Jihong 1 2 xingjihong2000@126.com
Dong Jingao 1 2 dongjingao@126.com
1 State Key Laboratory of North China Crop Improvement and Regulation, Hebei Agricultrual University , Baoding, Hebei , China
2 Hebei Key Laboratory of Plant Physiology and Molecular Pathology, Hebei Agricultural University , Baoding, Hebei , China
Wu Rongling
Electronic publication date: 2022 Nov 7
Publication date: 2022
Volume: 10
Electronic Location ID: e14306
Received 2022 Jun 21; Accepted 2022 Oct 5
Copyright: © 2022 Lv et al.
Copyright year: 2022
Copyright holder: Lv et al.
License: This is an open access article distributed under the terms of the Creative Commons Attribution License, which permits unrestricted use, distribution, reproduction and adaptation in any medium and for any purpose provided that it is properly attributed. For attribution, the original author(s), title, publication source (PeerJ) and either DOI or URL of the article must be cited.
License URL: https://creativecommons.org/licenses/by/4.0/

Keywords: Maize, NF-YA subunit gene, Biological stress, Abiotic stress, Expression analysis

Funding: National Natural Science Foundation of China 31901864 Key Research and Development Projects of Hebei 19226503D Central Government Guides Local Science and Technology Development Projects 216Z6501G, 216Z6502G Research Project of Basic Scientific Research Business Fees in Provincial Universities of Hebei Province KY2021043, KY2021044 China Agriculture Research System CARS-02 This work was supported by the National Natural Science Foundation of China (31901864), the Key Research and Development Projects of Hebei (19226503D), the Central Government Guides Local Science and Technology Development Projects (216Z6501G, 216Z6502G), the Research Project of Basic Scientific Research Business Fees in Provincial Universities of Hebei Province (KY2021043, KY2021044) and the China Agriculture Research System (CARS-02). The funders had no role in study design, data collection and analysis, decision to publish, or preparation of the manuscript.

==============================
NF-YAs encode subunits of the nuclear factor-Y (NF-Y) gene family. NF-YAs represent a kind of conservative transcription factor in plants and are involved in plant growth and development, as well as resistance to biotic and abiotic stress. In this study, 16 maize (Zea mays) NF-YA subunit genes were identified using bioinformatics methods, and they were divided into three categories by a phylogenetic analysis. A conserved domain analysis showed that most contained a CCAAT-binding transcription factor (CBFB) _NF-YA domain. Maize NF-YA subunit genes showed very obvious tissue expression characteristics. The expression level of the NF-YA subunit genes significantly changed under different abiotic stresses, including Fusarium graminearum infection and salicylic acid (SA) or jasmonic acid (JA) treatments. After inoculation with Setosphaeria turcica and Cochliobolus heterostrophus, the lesion areas of nfya01 and nfya06 were significantly larger than that of B73, indicating that ZmNFYA01 and ZmNFYA06 positively regulated maize disease resistance. ZmNFYA01 and ZmNFYA06 may regulated maize disease resistance by affecting the transcription levels of ZmPRs. Thus, NF-YA subunit genes played important roles in promoting maize growth and development and resistance to stress. The results laid a foundation for clarifying the functions and regulatory mechanisms of NF-YA subunit genes in maize.

Introduction

The nuclear factor-Y (NF-Y) transcription factor exists widely in eukaryotes and is also known as hemo-activator protein (HAP) (Thirumurugan et al., 2008; Petroni et al., 2012). NF-Y can bind to CCAAT-boxes in promoter sequences; consequently, it is also called CCAAT-binding factor (CBF) (Laloum et al., 2013). NF-Y is a large gene family composed of NF-YA (CBFB/HAP2), NF-YB (CBFA/HAP3) and NF-YC (CBFC/HAP5) subunits (Nardini et al., 2013). NF-Y is usually located in the nucleus and is evolutionarily conserved (Mantovani, 1999). In animals, NF-YA, NF-YB and NF-YC subunits are encoded by three single genes, and the three subunits function in the form of heterologous trimers (Benatti et al., 2008). In plants, the three subunits are encoded by more than 10 genes, and they can perform their functions independently (Petroni et al., 2012). The CBFB_NF-YA domain is the core conserved domain of the NF-YA family. The N-terminal of this domain can bind to NF-YB and NF-YC subunits, and the C-terminal can bind to DNA CCAAT-boxes (Quach et al., 2015). Compared with the rapid progress in mammalian and yeast (Saccharomyces cerevisiae) NF-Y protein-related research, research progress on the plant NF-Y family has been slow (Liang et al., 2012). Until now, this research has been limited to the preliminary bioinformatics comparisons between A. thaliana and related plant species, as well as the gene expression and function analyses.

The single subunit of NF-Y is widely involved in plant growth and development, such as controlling gametogenesis, embryo and plant development (Mu et al., 2013), abscisic acid (ABA) signal transduction (Yu et al., 2021), flowering cycle regulation (Hwang et al., 2019), primary root elongation (Ballif et al., 2011; Zhou et al., 2020), blue light response (Warpeha et al., 2007) and photosynthesis (Tokutsu et al., 2019) as well as stress responses, including to abiotic stresses such as drought, high temperature and salt (Li et al., 2021). The NF-YA subunit genes are involved in multiple processes in the plant lifecycle. In Arabidopsis thaliana, NF-YA3 and NF-YA8 genes mediate cell differentiation and embryo formation through the ABA signaling pathway during early embryonic development (Mu et al., 2013). NF-YA1, NF-YA5, NF-YA6 and NF-YA9 are involved in the development of gametes, embryos and seeds (Mu et al., 2013). AtNF-YA5 is regulated by miR169, thereby improving the resistance of Arabidopsis to drought stress (Li et al., 2008). The overexpression of OsNF-YA7 improves the drought tolerance of rice through an ABA-independent pathway (Lee et al., 2015). In potatos (Solanum tuberosum cv. ‘Desiree’), NF-YAs responds to drought by regulating the number of chlorophylls, stomatal conductance and photosynthesis (Li et al., 2021). PtNF-YA9 plays an important role in the drought resistance of Populus trichocarpa as a positive regulator of stress resistance (Lian et al., 2018). There are five NF-Y genes in tomato (Solanum lycopersicum) that play roles in tomato fruit ripening (Li et al., 2016). In soybean (Glycine max), GmNFYC14 forms heterotrimer with GmNF-YA16 and GmNFYB2, activates GmPYR1 mediated ABA signaling pathway and regulates soybean stress tolerance (Yu et al., 2021). In maize (Zea mays), ZmNFYB16 can form a heterotrimer with ZmNFYC17 and ZmNFYA01, and the heterotrimer binds to CCAAT cis-acting elements in the promoter region of stress response and growth-related genes through the ZmNFYA01 subunit, regulating the expression of multiple genes related to stress resistance and growth, thereby improving the drought resistance of plants (Wang et al., 2018). ZmNFYA03 promotes early flowering by binding to the FT-like12 promoter in maize (Su et al., 2018).

Plants have developed complex mechanisms to protect themselves against pathogens. Pathogenesis-related (PR) genes are the key elements of these mechanisms, and activated in response to pathogen attacks. They regulate production of several proteins, peptides or compounds which are toxic to pathogens or prevent pathogen infections where they start (Xie et al., 2010). The PR factors are thermostable, protease-resistant proteins of ~5–43 kDa which are expressed in all plant organs (Zribi, Ghorbel & Brini, 2021).

To date, an overall study of the maize NF-YA subunit gene family has not been reported, and the number, physicochemical properties and functions of maize NF-YA subunit genes are not clear. In this study, the maize NF-YA subunit genes were identified using bioinformatics methods, and their phylogenetic relationship, conserved domains, tissue specificity and gene expression patterns under biotic and abiotic stresses, such as salicylic acid (SA) and jasmonic acid (JA) treatments as well as disease resistance, were clarified. This would provide the foundation for elucidating the functions and regulatory mechanisms of maize NF-YA subunit genes.

Materials and Methods

Data sources and bioinformatics of NF-YA subunit genes in different species

The information and amino acid sequences of NF-YA subunit genes in maize, rice (Oryza sativa) and Arabidopsis were downloaded from MaizeGDB (http://www.maizegdb.org/), RGAP (http://rice.plantbiology.msu.edu/) and TAIR (https://www.arabidopsis.org/), respectively. The amino acid sequences were aligned using Clustal X software (Larkin et al., 2007). The aligned results were imported into MEGA 7.0 software (Kumar, Stecher & Tamura, 2016), and a phylogenetic tree was constructed using the maximum-likelihood method.

The chromosome location and the annotated information regarding the gene structure (including gene length, 5′-UTR, 3′-UTR and the distribution of each intron and exon) of maize NF-YA subunit genes were obtained from MaizeGDB, and their chromosome mapping was performed using RIdeogram software (https://cran.r-project.org/web/packages/RIdeogram/index.html). The gene structure map was drawn using IBS software (Liu et al., 2015).

The conserved domains of maize NF-YAs were analyzed using online software SMART (http://smart.embl-heidelberg.de/) and Pfam (http://pfam.xfam.org/), and the domain analysis maps were constructed using IBS software.

In accordance with the amino acid sequences of NF-YAs in maize, the subcellular localization was analyzed and predicted using Plant-mPLoc software (Chou & Shen, 2010).

For the predicted protein-protein interaction (PPI) analysis, all of the NF-YA amino acid sequences were searched using the STRING database version 11.5 (https://cn.string-db.org/) (Szklarczyk et al., 2021). A “confidence score” of STRING > 0.7 (high confidence) between proteins was used (Mei et al., 2021). The interaction networks of proteins generated using STRING were constructed to determine the relationships of proteins with NF-YAs.

Existing data-based expression patterns of NF-YA subunit genes in maize

Using the SRA database in NCBI (https://www.ncbi.nlm.nih.gov/), the RNA-seq data of 31 maize tissues (such as Seed_5_days_after_pollination, Endosperm_25_days_after_pollination, Seed_10_days_after_pollination, etc.) and those obtained under both abiotic stresses, such as high temperature (14 days of maize seedlings cultured at 50 °C for 4 h), low temperature (14 days of maize seedlings cultured at 5 °C for 16 h), salt (14 days of maize seedlings irrigated with 300 mM NaCl for 20 h), ultraviolet (14 days of maize seedlings irradiated by ultraviolet lamp for 2 h), drought (14 days of maize seedlings dried filter paper covered for 4 h) and Fusarium graminearum infection (200 μL of F. graminearum conidial suspension was inoculated onto the internodes of maize at ten-leaf stage after punctured with a pipette), were downloaded. Using Hisat2 software (http://daehwankimlab.github.io/hisat2/), RNA-seq data was aligned to the reference genome of maize. Using Cufflinks software (http://cole-trapnell-lab.github.io/cufflinks/), the gene expression value expressed as Fragments Per Kilobase of exon per Million fragments mapped reads (FPKM) was calculated using standardized parameters of gene length and number of reads. The expression heat map of the NF-YA family genes in maize was constructed using HemI software (Deng et al., 2014).

Plant materials and pathogenic fungal strains

The preserved seeds of maize inbred line B73, Setosphaeria turcica and Cochliobolus heterostrophus were from the Mycotoxin and Molecular Plant Pathology Laboratory, Hebei Agricultural University. F. graminearum strain PH-1 was provided by Prof. Mingguo Zhou at Nanjing Agricultural University. The ZmNFYA01 Mu insertion mutant nfya01 (Chr1, Insertion site 16041887, V4.0) and the ZmNFYA06 Mu insertion mutant nfya06 (Chr1, Insertion site 268308575, V4.0) were obtained from the ChinaMu Project (http://chinamu.jaas.ac.cn/) (Liang et al., 2019) and nfya01 and nfya06 plants used in this study were confirmed by PCR and quantitative real-time PCR (qRT-PCR) (Figs. S1 and S2). The primers for PCR and qRT-PCR identification are listed in Tables S1 and S2.

All the maize seeds were soaked in sterile water for approximately 24 h and potted in the mixture of vermiculite and nutrient soil at 1:1. The plants were cultured and grown in an artificial climate chamber with light for 14 h and darkness for 10 h at a temperature of 25–28 °C, and a humidity of 50–60%. The plants were irrigated once every 3 to 5 days, and the roots were irrigated with nutrient solution to ensure sufficient nutrition for the plants after reached the three-leaf stage.

S. turcica and C. heterostrophus were inoculated on Potato Dextrose Agar (PDA, 200 g/L potatos, 20 g/L glucose, 12 g/L agar) plates and grown in a 25 °C incubator.

F. graminearum was grown on PDA plates at 28 °C for 5 d, transferred to the carboxymethylcellulose sodium medium (CMC, 1.5 g/L CMC-Na, 1 g/L KH2PO4, 1 g/L NH4NO3, 1 g/L yeast powder and 0.5 g/L MgSO4·7H2O) and cultured in a shaker at 25 °C and 200 rpm for 5–7 d. The number of spores in the conidial suspension was counted using a blood cell counting plate, and more than 1 × 106 spores were inoculated onto maize.

Hormone treatments

In total, 1 L of salicylic acid (SA; 100 μM) and the same volume of jasmonic acid (JA; 100 μM) were sprayed independently and evenly on the aboveground parts of the two groups of maize B73 (five-leaf stage). Leaves were sampled at 0, 3, 9 and 24 h, frozen with liquid nitrogen and stored in a −80 °C refrigerator.

Leaf inoculation of maize

The 4th and 5th leaves of seven-leaf stage maize plants were cut off, and a wound of 1.2 cm in diameter was cut every 8 cm using a disposable syringe needle to facilitate the invasion of pathogenic fungi. Tween 20 was applied to the wound, and cultured S. turcica and C. heterostrophus were punched into the wound to cover the leaf wounds. Two layers of fully wetted filter paper were placed at the bottom of a white culture box. The leaves inoculated with pathogenic fungi were placed into the box, and covered with preservative film. The box had several vents in the box and was maintained in the darkness at 25 °C. The filter paper was kept moist, and the changes in the lesions were observed every day.

A total of 4 or 5 days after inoculation, the inoculated leaves were stained with trypan blue. The prepared 0.5% trypan blue staining solution and the leaves of an appropriate size were placed in a 50-mL centrifuge tube and boiled for 15 min to stain the necrotic cells. Then, 100 g chloral hydrate was dissolved into 40 mL water to make the chloral hydrate decolorizing solution. The stained leaves were washed with water to remove the dye solution of trypan blue, and then placed into chloral hydrate solution and shaken for 2 to 3 days to discoloring fully. Image J software (https://imagej.nih.gov/ij/) was used to measure the lesion area caused by fungal infection. The experiment was repeated three times. GraphPad Prism 8 software (https://www.graphpad.com/) was used to calculate the standard deviation (SD), and a Student’s t-test analysis was performed.

RNA extraction and qRT-PCR

A Plant RNA kit (OMEGA, Norcross, GA, USA) was used to extract sample RNAs, and a Reverse Transcription and cDNA Synthesis Kit (Clontech, Mountain View, CA, USA) was used to synthesize cDNA. Specific qRT-PCR primers for the internal reference gene UBQ9 (Jin et al., 2019) and maize NF-YA subunit genes (Table S2) synthesized by Beijing Bomede Biotechnology Co., Ltd. were used to analyze the expression of NF-YA subunit genes with two hormone treated maize inbred line B73 plant samples collected at different times as templates. The qRT-PCR primers of the internal reference gene UBQ1 (da Silva Santos et al., 2021) and ZmPRs (Table S3) were used for the expression analysis of ZmPRs with the leaf cDNAs of maize B73, nfya01 and nfya06 at the five-leaf stage as templates.

The reaction system was as follows: 7 μL of 2× M5 HiPer SYBR Premix EsTaq (with Tli RNaseH; TaKaRa, Dalian, China), 1 μL of cDNA template, 0.5 μL of forward primer, 0.5 μL of reverse primer and 5 μL of ddH2O. A fluorescence quantitative PCR instrument (CFX96 Real-time PCR Detection; BioRad, Hercules, CA, USA) was used for a total of 40 cycles, each of which was 95 °C for 30 s, 95 °C for 5 s and 60 °C for 30 s. Each qRT-PCR reaction was repeated three times, and the gene expression level was analyzed using the Ct value method (2−ΔΔCt). The SD of three replicates was calculated using GraphPad Prism 8 software, and the Student’s t-test analysis was performed.

Results

Bioinformatics analysis of maize NF-YA subunit genes

In total, 16 maize NF-YA subunit genes were obtained from MaizeGDB and named ZmNFYA01–16 in accordance with their chromosomal distribution. The 16 corresponding proteins differed in the number of amino acid (aa), relative molecular mass and isoelectric point. The lengths of the 16 NF-YA amino acid sequences were between 90 and 742 aa, with most being approximately 300 aa. The predicted isoelectric points indicated that most of these proteins were alkaline, with only the proteins encoded by ZmNFYA15 and ZmNFYA16 being acidic (Table 1). The 16 maize NF-YA subunit genes, 10 Arabidopsis NF-YA subunit genes and 11 rice NF-YA subunit genes could be divided into three groups: I, II and III. In maize, there were 6, 6 and 4 NF-YA subunit genes in I, II and III, respectively. Among them, ZmNFYA01 was orthologous to OsNFYA2, ZmNFYA06 was orthologous to OsNFYA5, ZmNFYA09 was orthologous to OsNFYA6, ZmNFYA10 was orthologous to OsNFYA4, and ZmNFYA14 was orthologous to OsNFYA1 (Fig. 1).

Table 1 Physicochemical properties of NF-YA subunit genes in maize.

Gene ID	Gene name	Chr	Start	End	AA	MW (Da)	pI	
Zm00001d027874	ZmNFYA01	1	16,042,002	16,038,734	249	27,207.16	8.96	
Zm00001d029489	ZmNFYA02	1	72,880,539	72,887,083	90	10,062.62	11.87	
Zm00001d031063	ZmNFYA03	1	175,269,072	175,263,789	742	85,041.59	9.42	
Zm00001d031092	ZmNFYA04	1	176,875,893	176,869,959	330	35,231.81	8.93	
Zm00001d033215	ZmNFYA05	1	254,416,539	254,420,845	322	34,122.58	9.61	
Zm00001d033602	ZmNFYA06	1	268,308,237	268,315,814	255	26,775.89	9.52	
Zm00001d033773	ZmNFYA07	1	273,183,116	273,187,001	262	29,076.26	9.80	
Zm00001d006835	ZmNFYA08	2	217,600,202	217,595,882	300	32,759.35	9.02	
Zm00001d007882	ZmNFYA09	2	241,675,483	241,672,970	294	31,336.04	9.78	
Zm00001d041491	ZmNFYA10	3	123,767,258	123,760,201	289	31,009.56	9.39	
Zm00001d013501	ZmNFYA11	5	12,927,450	12,924,588	126	14,415.92	10.90	
Zm00001d013676	ZmNFYA12	5	17,050,121	17,054,266	341	35,736.46	9.40	
Zm00001d013856	ZmNFYA13	5	22,800,236	22,795,868	156	16,876.48	9.64	
Zm00001d018255	ZmNFYA14	5	217,466,991	217,462,041	293	31,907.09	10.31	
Zm00001d022109	ZmNFYA15	7	170,564,261	170,559,870	271	29,607.50	6.63	
Zm00001d026305	ZmNFYA16	10	143,264,094	143,272,292	428	48,253.63	6.16	

Figure 1 Phylogenetic relationships of NF-YAs in Arabidopsis, rice and maize.

The maize NF-YA subunit genes were unevenly distributed among the chromosomes. Chromosome 1 contained the largest number (seven) of maize NF-YA subunit genes, ZmNFYA01–7. Chromosome 5 contained four maize NF-YA subunit genes: ZmNFYA11–14. Chromosome 2 contained two maize NF-YA subunit genes, ZmNFYA08 and ZmNFYA09. Chromosomes 3, 7 and 10 contained only one maize NF-YA subunit gene each, whereas chromosomes 4, 6, 8 and 9 did not contain any maize NF-YA subunit genes (Fig. S3). The lengths of the NF-YA subunit gene sequences were quite different, with ZmNFYA03, -07, -11 and -13 having no 5′-UTR and 3′-UTR structures. Introns existed in the 5′-UTRs of 10 genes, and most genes had 4–6 exons, suggesting that they may share the same RNA splicing pattern (Fig. S4).

Among the 16 members in the NF-YA family, 13 contained a CBFB_NF-YA domain, ZmNFYA03 contained a reverse transcriptase domain, and ZmNFYA13 had a TATA-binding protein domain. ZmNFYA07 lacked any known domains (Fig. 2A). The CBFB_NF-YA domain plays a key role in binding the NF-YB subunit and specifically binds the CCAAT box.

Figure 2 Conservative domain analysis and interaction protein network prediction of NF-YAs in maize.

(A) Phylogenetic analysis and domain analysis of NF-YAs in maize. Green box: CBFB_NF-YA domain; blue box: reverse transcriptase domain; red box: TATA-binding protein domain. (B) The PPI network of ZmNFYAs detected by STRING. Empty nodes: proteins of unknown 3D structure; filled nodes: some 3D structure is known or predicted.

ZmNFYA07 was predicted to be located in the chloroplast and cytoplasm, whereas ZmNFYA16 was predicted to be located in the mitochondrion and nucleus. The 14 remaining maize NF-YAs were predicted to be located in the nucleus (Table 2). Thus, most transcription factors were located in the nucleus, but they also played important roles in the mitochondrion and chloroplast.

Table 2 Subcellular localization prediction table for maize NF-YA subunit genes.

Gene name	Predicted location (s)	
ZmNFYA01	Nucleus	
ZmNFYA02	Nucleus	
ZmNFYA03	Nucleus	
ZmNFYA04	Nucleus	
ZmNFYA05	Nucleus	
ZmNFYA06	Nucleus	
ZmNFYA07	Chloroplast, Cytoplasm	
ZmNFYA08	Nucleus	
ZmNFYA09	Nucleus	
ZmNFYA10	Nucleus	
ZmNFYA11	Nucleus	
ZmNFYA12	Nucleus	
ZmNFYA13	Nucleus	
ZmNFYA14	Nucleus	
ZmNFYA15	Nucleus	
ZmNFYA16	Mitochondrion, Nucleus	

All NF-YA proteins were predicted to interact with proteins encoded by GRMZM2G180947_P01, GRMZM2G473152_P01 and GRMZM2G444073_P01. In addition, GRMZM2G143450_P01 and GRMZM2G099628_P01 may interact with ZmNFYA16. These two proteins may encode a methionine-tRNA ligase, which indicates that ZmNFYA16 may play a role in translation (Fig. 2B).

Existing data-based expression patterns of NF-YA subunit genes in maize

The expression levels of the 16 maize NF-YA subunit genes in the same tissues at different developmental stages were significantly different (P < 0.05). The expression of the same gene varied in different tissues at different stages. ZmNFYA01 was highly expressed at 25 d of embryonic development, and its expression levels in most tissues were significantly higher than those of the other genes, indicating that it played important roles in maize growth and development. ZmNFYA14 was highly expressed at 16 d and 25 d of embryonic development. ZmNFYA08 was highly expressed during embryonic development, endosperm development and seed germination. Low expression levels of ZmNFYA02, ZmNFYA07 and ZmNFYA13 were observed in all the tissues we studied (Fig. 3A).

Figure 3 Existing data-based expression patterns of NF-YA subunit genes in maize.

(A) Hierarchical clustering of expression levels of ZmNFYAs in 31 tissues. (B) Hierarchical clustering of expression levels of ZmNFYAs under abiotic stress. (C) Hierarchical clustering of expression levels of ZmNFYAs under biotic stress. Deep color indicates high expression; light color indicates low expression level.

The expression of ZmNFYA08 was significantly up-regulated under salt and drought stresses. The expression of ZmNFYA11 was similar under salt and drought treatments. ZmNFYA10 was up-regulated under heat and salt stresses. ZmNFYA02 was significantly down-regulated under heat, salt and UV stresses. The expression levels of ZmNFYA05, ZmNFYA12 and ZmNFYA14 decreased significantly under the heat stress. ZmNFYA07 expression was low and maintained without significant change under the various stresses (Fig. 3B).

The expression levels of ZmNFYA01 and ZmNFYA15 were the highest at 0 h after the fungal infection, and then decreased gradually. The expression levels of ZmNFYA02, -04–06, -08, -09, -11, -12 and -14 all showed a trend of first decreasing and then increasing. The expression of ZmNFYA03 and ZmNFYA10 increased first, then decreased and finally increased. The expression of ZmNFYA16 did not change significantly at 0–48 h after infection, but it decreased at 72 h after infection (Fig. 3C). Thus, the expression levels of NF-YA subunit genes in maize varied during F. graminearum infection, indicating that NF-YA subunit genes were involved and played important roles in maize disease resistance.

Expression of maize NF-YA subunit genes under hormone treatment

With the SA treatment, maize NF-YA subunit genes were divided into four categories based on their expression. In the first type, the expression increased first and then decreased. The expression levels of ZmNFYA01, -02 and -15 peaked at 3 h after treatment, and then decreased gradually. The expression levels of ZmNFYA03, -11, -13 and -14 peaked at 9 h after treatment and then decreased. In the second type, the expression decreased first and then increased. The expression levels of ZmNFYA04 and ZmNFYA05 were lowest at 9 h after treatment and then increased slightly. The expression of ZmNFYA12 was lowest at 3 h after treatment and then increased to the initial level. In the third type, the expression of the gene such as ZmNFYA08 and ZmNFYA16 continued to increase was maintained at a high level after the SA treatment. In the fourth type, the expression fluctuated after treatment. For example, the expression levels of ZmNFYA06 and ZmNFYA09 decreased first, then increased and finally decreased. The expression of ZmNFYA10 increased first and then decreased to its lowest level before increasing again (Fig. 4A).

Figure 4 Expression changes in ZmNFYAs following hormone treatments.

Expression changes after independent SA (A) and JA (B) treatments. Horizontal coordinates represent processing time. The experiments were repeated three times with similar results. Error bars indicate standard deviations. Asterisks indicate significant differences as assessed by Student’s t-tests (*P < 0.05; **P < 0.01).

Under the JA treatment, the maize NF-YA subunit genes were divided into three categories based on their expression patterns. In the first type, after the JA treatment, the gene expression peaked at 3 h after treatment and then gradually decreased, which included ZmNFYA01, -02, -04, -09, -12, -13, -15 and -16. In the second type, the expression increased first and then decreased before increasing again. The expression of ZmNFYA03, -06, -08 and -10 peaked at 3 h after treatment, decreased to their lowest levels at 9 h and then increased. In the third type, which included ZmNFYA05 and ZmNFYA11, expression decreased after the JA treatment. In addition, the expression of ZmNFYA14 decreased first and then increased before decreasing again (Fig. 4B). Thus, the NF-YA subunit genes may be involved and play important roles in SA and JA signaling pathways.

Positive regulation of maize disease resistance by ZmNFYA01 and ZmNFYA06

The lesion areas of nfya01 and nfya06 plants inoculated with S. turcica and C. heterostrophus were significantly larger than those of inbred line B73 (Figs. 5A–5D). Compared with B73, the sensitivity of nfya01 and nfya06 to S. turcica and C. heterostrophus was enhanced, indicating that ZmNFYA01 and ZmNFYA06 positively regulate the disease resistance of maize and provide broad-spectrum resistance to pathogenic fungi, thereby playing important roles in maize resistance to pathogen infection.

Figure 5 Resistance analyses of nfya01 and nfya06.

The leaves stained with trypan blue of B73, nfya01 and nfya06 inoculation with (A) S. turcica and (C) C. heterostrophus. (B and D) Measurement of the lesion areas on leaves shown in A and C, and the experiments were repeated three times with similar results. Error bars indicate standard deviations. Asterisks indicate significant differences as assessed by Student’s t-tests (*P < 0.05; **P < 0.01).

Expression analysis of ZmPRs in nfya01 and nfya06

Compared with in B73, the expression of ZmPR1–3, -5 and -6 in nfya01 and nfya06 plants were significantly down-regulated (P < 0.05), whereas the expression of ZmPR4, -7 and -10 were significantly up-regulated (P < 0.05) (Figs. 6A and 6B), indicating that ZmNFYA01 and ZmNFYA06 affected the expression of ZmPRs and suggesting that ZmNFYA01 and ZmNFYA06 participated in plant disease resistance by regulating the expression of ZmPRs. The expression patterns of different ZmPRs in nfya01 and nfya06 plants were consistent, indicating that ZmNFYA01 and ZmNFYA06 had similar functions.

Figure 6 Expression analyses of ZmPRs in nfya01 and nfya06.

Expression levels of ZmPRs in (A) nfya01 and (B) nfya06. The experiments were repeated three times with similar results. Error bars indicate standard deviations. Asterisks indicate significant differences as assessed by Student’s t-test (*P < 0.05; **P < 0.01).

Discussion

The roles of NF-YA subunit genes in the molecular mechanisms involved in maize responses to pathogen infection have been rarely reported. In this study, 16 NF-YA subunit genes were obtained by phylogenetic analysis and divided into three categories, with two more NF-YA members identified than what Zhang et al. (2016) have done. The eukaryotic 5′-UTR is critical for ribosome recruitment to the messenger RNA (mRNA) and in start codon choice. Additionally, it plays major roles in the control of translation efficiency and shaping the cellular proteome (Hinnebusch, Ivanov & Sonenberg, 2016). The 3′-UTRs of mRNAs regulate mRNA-based processes, such as mRNA localization, mRNA stability and translation. In addition, 3′-UTRs can establish 3′-UTR-mediated PPIs, and thus transmit genetic information encoded in 3′-UTRs to proteins. This function regulates diverse protein features, including protein complex formation or posttranslational modifications, but it may also alter protein conformations. Therefore, 3′-UTR-mediated information transfer can regulate protein features that are not encoded in the amino acid sequence (Mayr, 2019). However, ZmNFYA03, -07, -11 and -13 do not have 5′-UTR and 3′-UTR structures. Thus, further research is required to determine how they perform these functions.

The predicted PPI network indicated that all 16 NF-YA proteins interact with GRMZM2G180947_P01, GRMZM2G473152_P01 and GRMZM2G444073_P01 (Fig. 2B), which were named ZmNF-YB4 (GRMZM2G180947_P01), ZmNF-YB6 (GRMZM2G473152_P01) and ZmNF-YB9 (GRMZM2G444073_P01), respectively (Zhang et al., 2016). The naming is consistent with the binding of CBFB_NF-YA, a conserved domain in the associated NF-YA subunit genes, to the NF-YB subunit. Yang et al. (2022) confirmed that ZmNF-YB16 interacts with ZmNF-YC17 through its histone-folding domain to form a heterodimer in the cytoplasm. Then, the complex enters the nucleus under osmotic-stress conditions to form a heterotrimer with ZmNF-YA1 or ZmNF-YA7 (ZmNFYA08 in this study), forming the first identified ZmNF-Y transcriptional regulatory complex in maize (Yang et al., 2022). These results laid a foundation for elucidating the functions and regulatory mechanisms of maize NF-YA subunit genes. However, the specific functions of NF-YA subunit genes and their relationships with NF-YBs and NF-YCs still require further study.

The overexpression of ZmNF-YA1 enhances drought and salt tolerance and promotes root development in maize, whereas the zmnf-ya1 mutant shows drought and salt sensitivity (Yang et al., 2022). In this study, ZmNFYA01 was highly expressed during embryogenesis and positively regulated maize disease resistance, suggesting that it played important roles in maize growth and development. However, its regulatory mechanisms need further study.

Conclusions

In summary, 16 maize NF-YA subfamily genes were identified, and their expression levels in the same tissue at different developmental stages revealed a pattern, and the gene expression levels changed significantly under biotic and abiotic stress, including SA and JA treatments. ZmNFYA01 and ZmNFYA06 positively regulated maize resistance, and provided broad-spectrum resistance to pathogenic fungi. Compared with in B73, the expression levels of the ZmPRs in nfya01 and nfya06 plants were changed significantly, suggesting that this is part of the regulation of maize disease resistance. The important roles of NF-YA subunit genes in maize growth, development and resistance to biotic and abiotic stresses have been preliminarily determined.

Supplemental Information

Supplemental Information 1 PCR identification and qRT-PCR identification of nfya01.

(A) PCR identification, 1: nfya01F+nfya01R, 2: nfya01F+Mu67, 3: nfya01R+Mu67. (B) qRT-PCR identification. The experiments were repeated three times with similar results. Error bars indicate standard deviations. Asterisks indicate significant differences as assessed by Student’s t-tests (*P < 0.05; **P < 0.01).

Click here for additional data file.

Supplemental Information 2 PCR identification and qRT-PCR identification of nfya06.

(A) PCR identification, 1: nfya06F+nfya06R, 2: nfya06F+Mu67, 3: nfya06R+Mu67. (B) qRT-PCR identification. The experiments were repeated three times with similar results. Error bars indicate standard deviations. Asterisks indicate significant differences as assessed by Student’s t-tests (*P < 0.05; **P < 0.01).

Click here for additional data file.

Supplemental Information 3 Chromosomal localization of NF-YAs in maize.

Different genes are represented by dots of different colors. Stripes represent gene density.

Click here for additional data file.

Supplemental Information 4 Exon-intron organization of NF-YAs in maize.

Blue boxes: UTRs; green boxes: Exons; black lines: introns.

Click here for additional data file.

Supplemental Information 5 Primers for PCR identification of nfya01 and nfya06.

Click here for additional data file.

Supplemental Information 6 The qRT-PCR primers for maize NF-YA subunit genes.

Click here for additional data file.

Supplemental Information 7 The qRT-PCR primers for maize ZmPRs.

Click here for additional data file.

Supplemental Information 8 Raw data for gene expression.

Click here for additional data file.

Supplemental Information 9 Raw data for PRs expression.

Click here for additional data file.

Additional Information and Declarations

Competing Interests

Author Contributions

Data Availability

The authors declare that they have no competing interests.

Mingyue Lv conceived and designed the experiments, performed the experiments, analyzed the data, prepared figures and/or tables, authored or reviewed drafts of the article, and approved the final draft.

Hongzhe Cao conceived and designed the experiments, performed the experiments, analyzed the data, prepared figures and/or tables, authored or reviewed drafts of the article, and approved the final draft.

Xue Wang performed the experiments, analyzed the data, prepared figures and/or tables, and approved the final draft.

Kang Zhang performed the experiments, analyzed the data, prepared figures and/or tables, and approved the final draft.

Helong Si analyzed the data, prepared figures and/or tables, and approved the final draft.

Jinping Zang analyzed the data, prepared figures and/or tables, and approved the final draft.

Jihong Xing conceived and designed the experiments, authored or reviewed drafts of the article, and approved the final draft.

Jingao Dong conceived and designed the experiments, authored or reviewed drafts of the article, and approved the final draft.

The following information was supplied regarding data availability:

The raw data is available in the Supplemental Files.

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
