# Peer review of "Identification and expression analysis of maize NF-YA subunit genes"

_PeerJ, doi:10.7717/peerj.14306_

## Round 0.1 · original submission · Major Revisions

Both reviewers raised a number of issues that need to be addressed before I can accept the manuscript.

Reviewer 1 ·

Basic reporting

1. There are still some grammatical problems in the manuscript. It is suggested to seek the help of fluent speakers of English to polish the language of the manuscript. Such as line185-187.
2. The statistical results in the manuscript are lack of difference significance analysis. Please analyze the difference significance of these results.
3. In the result " Expression pattern of maize NF-YA subunit genes under abiotic stress ", the paragraph format is not indented by two characters in the first line, which is inconsistent with other paragraph formats.
4. Line 421: “ZmNFYAs” is not italicized. Please carefully check the full text and modify it.
5. In Figure 5, there are two " leaves 20 day old seeding ". Please check for errors and change them.
6. "Maize" is used in the full text, but "Zea mays" is used in the keywords. Please change the keywords.
7. Some references are not in the same format as other documents, please unify the format.

Experimental design

no comment

Validity of the findings

In the manuscript, the author identified maize NF-YA subunit gene by bioinformatics method, and studied the expression pattern of NF-YA subunit gene under biological stress, abiotic stress, SA and JA treatment, which laid a foundation for clarifying the function of NF-YA subunit gene in maize. However, the manuscript has the following problems:
1. The manuscript only studied the expression patterns of NF-YA subunit gene in salicylic acid, jasmonic acid, biological stress and abiotic stress, and concluded that NF-YA subunit gene plays an important role in maize, which is manifested in promoting maize growth and development and stress resistance. I don't think these data are enough to support this result. I suggest supplementing relevant experiments to make the manuscript more convincing.
2. When studying the expression pattern of NF-YA subunit gene under abiotic stress, what are the conditions of heat, cold, salt, UV and drought? For example, what is the temperature? What is the salt concentration? Please list in materials and methods.
3. When studying the expression pattern of NF-YA subunit gene under biological stress, what are the growth conditions of Fusarium graminearum? Under what conditions can corn be infected? What is the mode of infection? Please list in materials and methods.
4. In this study, what are the growth conditions and environment of experimental material maize? How long after growth can it be used for a series of experiments?
5. ZmNFYA03、ZmNFYA07、ZmNFYA11 and other genes lack CBFB\UNF-YA domain. In the introduction, it is introduced that CBFB\UNF-YA domain has a lot of correlation with its function, so ZmNFYA03、ZmNFYA07、ZmNFYA11 and other genes have other domains to play a role in driving function?
6. There are many contents about the gene structure of NF-YA subunit in the result part of the manuscript, and the content about the correlation between these structures and gene function can be added in the discussion part.

Reviewer 2 ·

Basic reporting

1. Line 60: please add references to the end of the sentence.
2. It is suggested to add Latin names of organisms when they first appear in the manuscript, such as potato, tomato, and rice. In some sentences cited from other references, maybe some variety or cultivar or clone is used and included.
3. In Plant materials, it is suggested to add a brief description of seedling culture since people who don’t specialize in maize don’t understand what V5 stage (L118) refers to. What is more, the specific tissue used in the experiments should be mentioned.
4. L105: it is suggested to change “Expression pattern of NF-YA subunit genes in maize” to “Existing data-based expression patterns of NF-YA subunit genes in maize”, the latter of which could be distinguished from “Expression analysis of NF-YA subunit genes in maize under hormone treatments”.
5. L118: what is V5 stage? Which tissues were used in the experiment?
6. L118-119: the sentence is not complete. What is the rationale for the sampling strategy (time)?
7. L127: pl. add the producer and model to the RT-PCR instrument. The sentence is not complete.
8. The first sentence of each part in the Results is a repeated description of methods, which is unnecessary and suggested to delete.
9. L174-177: it is not good to have two attributive clauses in one sentence. It is suggested to reorganize the sentence.
10. L179-211: it is suggested to have a heading for these paragraphs, which correspond to “Expression pattern of NF-YA subunit genes in maize” in the Methods, and the headings of these paragraphs could be used as subheadings.
11. In Discussion, it is unnecessary to repeat the results previously described. Comparisons between your results and those from others and some explanations could be made if possible.
12. In Conclusions, conclusions should be directly described. It is unnecessary to describe what has been performed.
13. It is suggested to turn to a fluent English speaker to modify the manuscript in terms of English language although the reviewer has made some corrections (pl. read the manuscript for details).

Experimental design

Pl. see above for details.

Validity of the findings

Pl. see above for details.

Additional comments

No.

Annotated reviews are not available for download in order to protect the identity of reviewers who chose to remain anonymous.

---

## Round 0.2 · Minor Revisions

Please make a final revision according to reviewer 1's suggestions.

Reviewer 1 ·

Basic reporting

After careful inspection of the author response and other reivewer's comments, I feel this manuscript is ready for publication at current form.

Experimental design

no comment

Validity of the findings

no comment

Reviewer 2 ·

Basic reporting

1. It is suggested to move Lines (L) 136 and 137 to the end of the sentence on L129. Why is Fusarium graminearum mentioned here since no relevant description is found in Results?
2. L145-149: please see the comment No. 1.
3. L184: no description of ZmPRs is described prior to L184. Pl. add some description.
4. L242: please specify “all the stages”.
5. The first paragraph in Discussion is too long. Please have it divided into paragraphs.
6. L298-303 and L331-335: it would be more appropriate to put these sentences in Introduction than in Discussion.
7. In Conclusions, only should main results of the study be mentioned and repeated.
8. Please read the manuscript reviewed for detailed suggestions for modifications.

Experimental design

Pl. see above.

Validity of the findings

Pl. see above.

Additional comments

No comment.

Annotated reviews are not available for download in order to protect the identity of reviewers who chose to remain anonymous.

---

## Round 0.3 · accepted · Accept

The authors have satisfactorily addressed the reviewers' comments.